# Assessing the Perceptions and Impact of Critical Incident Stress Management Peer Support among Firefighters and Paramedics in Canada

**DOI:** 10.3390/ijerph19094976

**Published:** 2022-04-20

**Authors:** Jill A. B. Price, Caeleigh A. Landry, Jeff Sych, Malcolm McNeill, Andrea M. Stelnicki, Aleiia J. N. Asmundson, R. Nicholas Carleton

**Affiliations:** 1Department of Psychology, University of Regina, Regina, SK S4S 0A2, Canada; caeleigh.landry@uregina.ca (C.A.L.); andrea.stelnicki@gmail.com (A.M.S.); nick.carleton@uregina.ca (R.N.C.); 2Sych & Associates Psychological Services, Edmonton, AB T5M 2P6, Canada; jeffsych@shaw.ca (J.S.); malcolm.psych@shaw.ca (M.M.); 3Department of Psychology, University of Calgary, Calgary, AB T2N 1N4, Canada; aleiia.asmundson@ucalgary.ca

**Keywords:** Critical Incident Stress Management, peer support, mental disorders, mental health, public safety personnel

## Abstract

Relative to the general population, public safety personnel (PSP) appear at an increased risk of developing mental health challenges as a result of repeated exposure to potentially psychologically traumatic events (PPTEs). To help mitigate the impact of PPTEs on PSP mental health, many PSP agencies have implemented diverse peer support despite limited empirical evidence. The current study was designed to expand the empirical evidence surrounding peer support by investigating one of the most widely used and structured peer support resources: Critical Incident Stress Management (CISM). Specifically, the current study with integrated firefighters and paramedics assessed (a) the prevalence of mental disorders; (b) perceptions of high fidelity CISM peer support; and (c) the comparative associations of CISM with high fidelity (*n* = 91) versus unknown fidelity (*n* = 60) versus no CISM (*n* = 64) and mental health. Results indicated that (a) mental disorders are prevalent among PSP irrespective of gender, age, and years of service; (b) participants perceived CISM peer support as offering beneficial and valuable tools (e.g., skills and coping strategies); and (c) high fidelity CISM environments offer some mental health benefits to individuals who screen positive for alcohol use disorder and generalized anxiety disorder. Overall, the current study offers novel information that can inform future directions for evidence-based peer support and policy decisions designed to support the mental health of PSP.

## 1. Introduction

The work of public safety personnel (PSP; e.g., border security personnel, correctional workers, firefighters, police, paramedics, public safety communicators) consists of unique demands and stressors that can increase the risk of exposure to potentially psychologically traumatic events (PPTEs) [1]. A PPTE involves actual or threatened death, serious injury, or sexual violence experienced by the self or others (e.g., serious transportation accident, sexual assault, combat) [2,3]. Particularly debilitating PPTEs may include unexpected death or gruesome injuries particularly among vulnerable victims (e.g., children) [4,5]. Approximately 90% of the general population report exposure to at least one PPTE in their lifetime [6]; in contrast, PSP report being exposed to hundreds or even thousands of PPTEs throughout their career, increasing their risk of several mental disorders [1,2].

### 1.1. Mental Health

Prevalence of positive screens for a mental disorder among PSP vary by individual differences. PSP who are older or have served more years are more likely to screen positive for one or more mental disorder [1,7]. PSP from Western Canada appear more likely to screen positive for a mental disorder than PSP from Eastern Canada [7]. Women PSP appear more likely to screen positive for a mental disorder than men PSP; however, the difference must be interpretated with caution as women may be more likely to report experiencing symptoms because of gender differences in stigma [7,8]. Mental disorders commonly reported among PSP include alcohol use disorder, generalized anxiety disorder, major depressive disorder, panic disorder, posttraumatic stress disorder (PTSD), and social anxiety disorder [7]. The odds of PSP screening positive for a mental disorder increase with exposure to most PPTEs, particularly severe human suffering, physical assault, or sudden violent death [1]. Positive screens for social anxiety disorder have also been associated with sexual assault, other unwanted or uncomfortable sexual experience, or captivity [1]. Various mental health resource provisions (e.g., Critical Incident Stress Management [CISM], Critical Incident Stress Debriefing, Mental Health First Aid, Peer Support, Road to Mental Readiness) have been associated with decreased odds of screening positive for PTSD, major depressive disorder, generalized anxiety disorder, and social anxiety disorder among PSP [9]; however, to date, there appears to be little association between such mental health resources and positive screens for panic disorder or alcohol use disorder among PSP [9].

### 1.2. Peer Support

Mental disorders represent a significant health concern and economic burden for PSP, as well as the organizations and communities that depend on their support [10,11]. To help mitigate the impact of stressful work environments, many PSP agencies (e.g., fire departments) have implemented peer support. Peer support refers to a supportive relationship between peers who have a lived common experience [12]. Potential advantages of peer support for PSP include: (1) talking to individuals who are familiar with and understand the unique demands of PSP work; (2) greater accessibility relative to formal mental health resources; and (3) reduced power imbalance relative to what might be perceived with a registered mental health care provider [13,14]. Perceived social support from peers can also have a positive impact on resiliency [15] as well as cumulative stress (e.g., decreased socialization, interpersonal relationship problems, increased irritability, anger, aggression) [16,17]. Despite the promotion within PSP agencies, the empirical evidence surrounding peer support remains insufficient to broadly inform policy decisions designed to protect PSP mental health [18,19].

### 1.3. CISM

Developed by the International Critical Incident Stress Foundation, CISM is one of the most widely used and structured peer support resources [18,19]. CISM provides detailed, integrative, and multi-phase peer support that: (1) incorporates specific tools tailored for psychological injury and stress (i.e., resistance, resilience, recovery); (2) emphasizes the value in the peer relationship to reconnect individuals to their adaptive coping strategies; and (3) fosters group cohesion, performance, and social connectedness. Support can include providing any number of training courses based on the needs of any given PSP person or organization. There is an Approved Instructor Candidate Training Program for CISM that involves completing prerequisite advanced courses, an adult instructor education course, and course-specific training (e.g., Approved Instructor Training course for Assisting Individuals in Crisis). Upon successful completion, the Approved Instructor is required to sign an agreement to provide the course instruction with high fidelity. Fidelity refers to the extent to which delivery adheres to the model originally developed. Specifically, an Approved CISM instructor is restricted to no more than 10% deviation to meet the specific cultural and demographic needs of a given trainee population. Full certification requires training in at least Assisting Individual in Crisis and Group Crisis Intervention, which requires 32 h of instruction [20]. Despite the extensive training requirements, there has been little research on the delivery of CISM peer support among PSP across different levels of fidelity and associations with mental health [18,19].

### 1.4. Research Questions and Hypotheses

The current study was designed to provide insights into the perceptions and impact of CISM peer support among integrated firefighters and paramedics in Canada. Integrated means that each participant is a trained firefighter who is also certified as either a primary care paramedic or an advanced care paramedic. Specifically, the current study assessed whether (a) prevalence of mental disorders (i.e., alcohol use disorder, generalized anxiety disorder, major depressive disorder, panic disorder, PTSD, social anxiety disorder) among CISM peer support participants varied by individual differences (i.e., gender, age, years of service); (b) perceptions of high fidelity CISM peer support varied by individual differences and screens for mental disorders; and (c) screens for mental disorders varied by CISM peer support experience (i.e., CISM with high fidelity, CISM with unknown fidelity, and no CISM). The research team hypothesized that (a) screens for mental disorders would not vary by individual differences; (b) perceptions of high fidelity CISM peer support would not vary by individual differences or screens for mental disorders; and (c) participants who received high fidelity CISM peer support would report fewer symptoms of mental disorders relative to a control group of participants who reported receiving CISM peer support of unknown fidelity, and relative to a control group of participants who reported having no CISM experience.

## 2. Materials and Methods

### 2.1. Participants

Using a quasi-experimental design, participants (*n* = 91) for the experimental group were recruited from four integrated Fire and Emergency Medical Services departments in Alberta, Canada, all of which offer high fidelity CISM peer support to their members. The experimental group participants (*n* = 91) predominantly self-identified as men (93.3%), between 30 and 39 years of age (50.5%), with 10 to 19 years of experience working within public safety (45.1%). Participants for the control group were derived using data from an extant sample [7]. To maintain sample consistency with the experimental group, participant data for the control group were included only for PSP who self-identified as being both a firefighter and paramedic. Approximately half of the control participants reported previously receiving CISM peer support of unknown levels of fidelity (*n* = 60) and approximately half reported receiving no CISM peer support at all (*n* = 64). The control group participants who reported receiving CISM peer support of unknown fidelity predominantly self-identified as men (91.7%), between 30 and 39 years of age (31.7%), with 10 to 19 years of experience working within public safety (43.3%). The control participants who reported no CISM peer support experience predominantly self-identified as men (71.9%), between 30 and 39 years of age (31.3%), with 10 to 19 years of experience working within public safety (35.9%). Ethics approval was obtained from the first author’s University Institutional Research Ethics Board.

### 2.2. Procedure

#### 2.2.1. Experimental Group

Department eligibility was based on five criteria. First, all departments had to be in Alberta, Canada. Second, all departments had to offer CISM peer support. Third, the department CISM peer supporters must have received high fidelity training as well as ongoing refresher training from Approved Instructors. Fourth, the department CISM peer supporters had to have demonstrably incorporated all six core CISM components into their processes (i.e., assessment and triage; strategic planning; individual crisis intervention; informational group crisis intervention; interactive group crisis intervention; resiliency). Fifth, each department had to have access to the same clinical director; specifically, a registered mental health care provider in Alberta who was trained in CISM peer support and attends quarterly CISM team meetings to review peer support activities as well as monitor fidelity. Participant eligibility was based on two criteria. First, participants had to be currently deployed as a PSP at one of the eligible Fire and Emergency Medical Services departments in Alberta, Canada. Second, each participant had to be an integrated firefighter and paramedic.

Eligible participants were provided with a recruitment package that included a consent form, information on the procedures, goals of the study, and self-report measures. The consent form explained the voluntary nature of participation and that participants could dispose of the recruitment package at any time. Participants who initially completed the self-report measures and later decided to discontinue could either confidentially dispose of the recruitment package themselves or make a note for the research team to confidentially dispose of the package once returned. In total, the self-report measures took approximately 15 mins to complete. Participants were instructed that they had 75 days, from when the last set of recruitment packages were distributed at their department, to complete and return the consent form and self-report measures. The recruitment package included a pre-addressed, pre-paid postage envelope so that participants could return their responses anonymously and at their own convenience. A research assistant was available to provide more information about the study and to answer any questions.

#### 2.2.2. Control Participants

Participant eligibility was based on two criteria. First, participants had to be derived using data from an extant sample [7]. Second, each participant had to be an integrated firefighter and paramedic. Eligible participants were originally recruited via emails (see [7] for details). Participants were provided with an online survey link that directed them to a consent form, information on the procedures, goals of the study, and self-report measures. The consent form explained the voluntary nature of participation and that participants could stop completing the online assessments at any time. Participants were encouraged to reach out to the research team via the contact information provided with any questions or concerns. Participants could not request their information be withdrawn from the study after clicking submit because individual participants could not be identified.

### 2.3. Measures

#### 2.3.1. Demographics Survey

The demographic survey consisted of 18 items that included both general and specific questions tailored for PSP. General questions targeted participants’ age, gender, religious beliefs, marital status, household composition, family income, and level of education, whereas specific questions inquired about participants’ employment field, vacation days, sick days, length of time in their current position, time spent in front-line duty, and the age at which they began working as a PSP. The demographic survey was designed to properly define the population sample.

#### 2.3.2. Peer Support Survey

The Peer Support Survey is a 24-item self-report measure modified from the Resilience Training Evaluation [21]. The Peer Support Survey assessed participants’ perceived strengths and weaknesses on four dimensions of peer support (i.e., skills, use, value, impact) rated on five-point Likert-type scales. The skills section consisted of 5 items that were used to measure participants’ perceived skills acquired via CISM peer support from 1 (strongly disagree) to 5 (strongly agree). The use section consisted of 5 items that measured participants’ perceived use of five coping strategies (i.e., mental rehearsal, self-talk, tactical breathing, goal setting, attention control) from 1 (never) to 5 (regular basis). The value section consisted of 6 items that measured participants’ perceived value of CISM peer support from 1 (strongly disagree) to 5 (strongly agree). The impact section consisted of 5 items rated on a three-point Likert-type scale and measured participants’ perceived effectiveness of the coping strategies acquired from CISM peer support from 0 (made things worse) to 2 (made things better). The Peer Support Survey demonstrated strong internal consistency (α = 0.91).

#### 2.3.3. Alcohol Use Disorders Identification Test (AUDIT)

The AUDIT is a 10-item self-report measure used to assess symptoms of alcohol use disorder based on the Diagnostic and Statistical Manual of Mental Disorders, 5th Edition (DSM-5) [22]. Each item was rated on a five-point Likert-type scale ranging from 0 (low frequency and use) to 4 (high frequency and use) and included questions such as “How often do you have six or more drinks on one occasion?”. The AUDIT demonstrated strong internal consistency (α = 0.77). A positive screen for alcohol use disorder was coded if participants reported a total score greater than 15 [22].

#### 2.3.4. Generalized Anxiety Disorder (GAD-7)

The GAD-7 scale is a 7-item measure used to assess the frequency of generalized anxiety disorder symptoms over the past two weeks [23]. Each item was rated on a five-point Likert-type scale ranging from 0 (not at all) to 3 (nearly every day) and included questions such as “Over the past 2 weeks, how often have you been bothered by [feeling nervous, anxious or on edge]?”. The GAD-7 also demonstrated strong internal consistency (α = 0.89). A positive screen for generalized anxiety disorder was coded if participants reported a total score greater than 10 [23].

#### 2.3.5. Panic Disorder Severity Scale, Self-Report (PDSS-SR)

The PDSS-SR is a 7-item measure designed to assess the frequency and severity of panic disorder symptoms based on the DSM-5 [24]. Each item was rated on a five-point Likert-type scale ranging from 0 (no interference) to 4 (extreme, incapacitating impairment) and included questions such as “How many panic attacks did you have during the week?”. The PDSS-SR demonstrated strong internal consistency (α = 0.91). A positive screen for panic disorder was coded if participants reported a total score greater than 7 [24,25].

#### 2.3.6. Patient Health Questionnaire (PHQ-9)

The PHQ-9 is a 9-item self-report measure used to assess the severity of major depressive disorder symptoms over the last two weeks [26]. Each item was rated on a five-point Likert-type scale ranging from 0 (not at all) to 3 (nearly every day) and included questions such as “Over the past 2 weeks, how often have you been bothered by feeling down, depressed, or hopeless?”. The PHQ-9 also demonstrated strong internal consistency (α = 0.87). A positive screen for major depressive disorder was coded if participants reported a total score greater than 9 [27].

#### 2.3.7. Posttraumatic Stress Disorder Checklist (PCL-5)

The PCL-5 [28] is a 20-item scale designed to assess four clusters of PTSD symptoms based on the DSM-5: intrusive thoughts, avoidance, negative alterations in cognition and mood, and alterations in arousal and reactivity [2]. Each item was rated on a five-point Likert-type scale ranging from 0 (not at all) to 4 (extremely) and included questions such as “In the past month, how much were you bothered by repeated, disturbing, and unwanted memories of the stressful experience?” [28]. The PCL-5 demonstrated strong internal consistency (α = 0.92). A positive screen for PTSD was coded if participants met the minimum criteria for each DSM-5 symptom cluster and exceeded a total score of 32 [28].

#### 2.3.8. Social Interaction Phobia Scale (SIPS)

The SIPS is a 14-item measure designed to assess the symptoms of social anxiety disorder, using three DSM-5 dimensions: social interaction anxiety, fear of overt evaluation, and fear of attracting attention [2,29]. Each item was rated on a five-point Likert-type scale ranging from 0 (not at all characteristic of me) to 4 (entirely characteristic of me) and included questions such as “I have difficulty talking with other people”. The SIPS also demonstrated strong internal consistency (α = 0.92). A positive screen for social anxiety disorder was coded if participants reported a total score greater than 20 [29].

### 2.4. Statistical Analyses

Descriptive statistics were conducted to summarize the individual differences and positive screens for mental disorders across participant groups. Binary logistic regression analyses were conducted to compare screens for mental disorders by individual differences. Multiple regression analyses were conducted to assess the level of variances in participants’ perceptions of high fidelity CISM peer support by individual differences and screens for mental disorders. Lastly, binary logistic regression analyses were conducted to compare screens for mental disorders based on CISM experience.

## 3. Results

Descriptive statistics were calculated to summarize the individual differences and positive screens for mental disorders across three different CISM environments (i.e., high fidelity, unknown fidelity, no CISM experience; see Table 1). Participants consisted predominantly of men who were employed full-time as integrated firefighters and paramedics. A total of 36.7% participants in the experimental group screened positive for at least one mental disorder. The frequency of positive screens was more than triple of the general population (i.e., 10.1%) [30]; however, the positive screens for at least one mental disorder were comparable to a general sample of firefighters (34.1%) and less than a general sample of paramedics (49.1%) [7]. Due to a logistical design error in the Peer Support Survey, perceived effectiveness of coping strategies acquired via CISM peer support was not included in any statistical analyses because fewer than 25% of participants completed the necessary questions. Participants in the control sample did not complete the Peer Support Survey; as such, perceptions of CISM peer support were only calculated for participants in the experimental group.

Binary logistic regression analyses were conducted to compare screens for mental disorders by individual differences (i.e., gender, age, years of service). Younger participants were statistically significantly more likely to screen positive for alcohol use disorder than older participants, Exp(B) = 0.88, CI [0.80, 0.98], *p* = 0.018. There were no statistically significant differences in alcohol use disorder screens based on gender or years of service. Similarly, there were no statistically significant differences between screens for generalized anxiety disorder, major depressive disorder, panic disorder, PTSD, or social anxiety disorder based on gender, age, or years of service.

### 3.1. Perceptions of CISM Peer Support

#### 3.1.1. Individual Differences

Multiple regression analyses were conducted to assess the level of variances in participants’ perceptions of high fidelity CISM peer support by individual differences (i.e., gender, age, years of service). Participants’ perceived skills acquired via CISM did not vary statistically significantly by individual differences, *F*(4, 60) = 1.56, *p* = 0.198, R^2^ = 0.094. Perceived use of coping strategies (i.e., mental rehearsal, self-talk, tactical breathing, goal setting, attention control) acquired via CISM also did not vary statistically significantly by individual differences, *F*(4, 59) = 0.78, *p* = 0.543, R^2^ = 0.050. Similarly, participants’ perceived value of CISM did not vary statistically significantly by individual differences, *F*(4, 57) = 2.18, *p* = 0.082, R^2^ = 0.133.

#### 3.1.2. Mental Disorders

Multiple regression analyses were conducted to assess the level of variances in participants’ perceptions of high fidelity CISM peer support by screens for mental disorders (i.e., alcohol use disorder, generalized anxiety disorder, major depressive disorder, panic disorder, PTSD, social anxiety disorder). Perceived peer support skills varied statistically significantly by mental disorder screens, *F*(6, 57) = 3.33, *p* = 0.007, R^2^ = 0.259; however, PTSD, *t* = −2.14, *p* = 0.037, was the only statistically significant predictor in the model. Participants who screened positive for PTSD reported acquiring statistically significantly fewer perceived CISM skills. Alcohol use disorder, generalized anxiety disorder, major depressive disorder, panic disorder, and social anxiety disorder were not statistically significantly associated with perceived CISM skills. Perceived use of CISM coping strategies (i.e., mental rehearsal, self-talk, tactical breathing, goal setting, attention control), *F*(6, 57) = 0.99, *p* = 0.443, R^2^ = 0.094, and perceived value of CISM, *F*(6, 54) = 1.79, *p* = 0.118, R^2^ = 0.116, did not vary statistically significantly by mental disorder screens.

#### 3.1.3. Impact of CISM Peer Support

Binary logistic regression analyses were conducted to compare screens for mental disorders based on CISM peer support experience (i.e., CISM with high fidelity, CISM with unknown fidelity, no CISM). Participants in high fidelity CISM environments were statistically significantly less likely to screen positive for alcohol use disorder, Exp(B) = 0.30, CI [0.11, 0.80], *p* = 0.017, and generalized anxiety disorder, Exp(B) = 0.29, CI [0.10, 0.82], *p* = 0.020, than participants who reported no CISM experience. Participants in high fidelity CISM environments screened positive for major depressive disorder, panic disorder, PTSD, and social anxiety disorder at rates comparable to participants who reported no CISM experience. Participants in the unknown fidelity CISM environments screened positive for alcohol use disorder, generalized anxiety disorder, major depressive disorder, panic disorder, PTSD, and social anxiety disorder at rates comparable to participants who reported no CISM experience.

## 4. Discussion

The current study investigated the perceptions and impact of CISM peer support among integrated firefighters and paramedics in Canada. The first set of research questions explored the prevalence of mental disorders. Results support research showing evidence that screens for alcohol use disorder, generalized anxiety disorder, major depressive disorder, panic disorder, PTSD, and social anxiety disorder are prevalent among PSP [7]. Many participants in the current study screened positive for at least one mental disorder (36.7%), which appears more than three times of the diagnostic rate from the general population (i.e., 10.1%) [30], but less than that of a general sample of diverse PSP (44.5%) [7]. The discrepant prevalence between the studies may be explained by differences in the samples. The pan-Canadian study included a relatively equal distribution of participants from each PSP sector [7]; in contrast, the current sample was exclusive to integrated firefighters and paramedics. Nevertheless, the current sample (36.7%) falls within the prevalence rates of firefighters (34.1%) and paramedics (49.1%) [7].

Mental disorders are also prevalent among PSP irrespective of individual differences. The current study provides evidence that gender, age, and years of service did not statistically significantly impact screens for generalized anxiety disorder, major depressive disorder, panic disorder, PTSD, or social anxiety disorder. Gender and years of service also did not influence screens for alcohol use disorder; however, consistent with previous research [31], younger participants were statistically significantly more likely to screen positive for alcohol use disorder than older participants. To help promote mental health, many PSP agencies are offering supplemental resources (e.g., CISM). The current study provides novel information on the perceptions and impact of CISM peer support.

### 4.1. Perceptions of CISM Peer Support

The second set of research questions explored perceptions of CISM peer support. Investigating perceptions of peer support can help identify potential peer support benefits. Peer support was evaluated via perceived skills acquired, use of coping strategies, and value of CISM. Participants perceived their experience of CISM as beneficial irrespective of individual differences. Specifically, the current study provides evidence that gender, age, and years of service did not statistically significantly impact perceived skills acquired, use of coping strategies, or value of CISM.

The current study found evidence that participants who screened positive for PTSD reported having acquired statistically significantly fewer perceived mental health skills (e.g., mindfulness). The cognitive demands of certain mental health challenges (e.g., PTSD) may hamper potential benefits of mental health services, including CISM. The current results may underscore the importance of including registered mental health care providers among peer support teams, as well as the importance of clear standard operating procedures for stepped-care referrals (e.g., psychotherapy). Mental health challenges did not appear to hinder participants’ use of coping strategies and tools facilitated by CISM, or the perception of CISM as beneficial.

### 4.2. Impact of CISM Peer Support

The third set of research questions explored the impact of fidelity in CISM peer support experiences. Understanding the relationship between fidelity and effectiveness is important given the contemporary diversity of peer support programming [18]. Despite evidence that fidelity is inversely related to beneficent outcomes [32], there is substantial variability in fidelity of peer support and other mental health resource provisions across Canada [18,19]. The current study helps inform the relationship between fidelity and peer support by comparing three different environments: CISM peer support delivered with high fidelity, CISM peer support delivered with unknown fidelity, and no CISM peer support experience. Participants who received CISM with high fidelity were statistically significantly less likely to screen positive for alcohol use disorder and generalized anxiety disorder than participants with no CISM experience. Participants who received CISM with high fidelity screened positive for major depressive disorder, panic disorder, PTSD, and social anxiety disorder at rates comparable to participants who received CISM with unknown fidelity or who reported no CISM experience. Participants who received CISM with unknown fidelity screened positive for alcohol use disorder and generalized anxiety disorder at rates comparable to participants who reported no CISM experience.

CISM may be providing some benefits for symptoms of alcohol use disorder and generalized anxiety disorder when delivered with high fidelity. Peer support appears to influence drinking rates, barriers to seeking substance use treatment, and ongoing sobriety [33]. Peer support appears to also serve as a protective factor against intolerance of uncertainty [34] and facilitate psychological safety [35], which may help to minimize symptoms of generalized anxiety disorder. The variability of benefits underscores the need for stepped care options across diverse PSP environments. The results also suggest that fidelity may play an important critical role in the delivery of CISM, and future research is recommended to explore the impact high fidelity CISM environments on mental disorders using a longitudinal design.

### 4.3. Limitations

The current study has limitations that can inform directions for future research. First, the current study used a quasi-experimental design in which the control and experimental groups were collected via different means. Participants were not randomly assigned to their group, which means that the samples may be unintentionally biased. The sample also cannot infer causality. Future researchers are encouraged to use a longitudinal design with random controlled trials to evaluate the benefits of peer support before and after participation. Second, the sample consisted of predominantly men, limiting generalizability of results related to gender. Third, the experimental data is geographically restricted to Alberta and derived from integrated firefighters and paramedics. Research evaluating larger geographic regions, as well as the impact of peer support in other PSP sectors, would better inform the generalizability of potential peer support benefits. Fourth, only the experimental group completed the Peer Support Survey. Future research should ensure that both the experimental and control groups complete all measures. Fifth, the Peer Support Survey is a relatively new tool; as such, the tool may not have been sufficiently nuanced to measure the perceived impact of skills, use of coping strategies, or value of CISM. Future researchers should continue to evaluate the psychometric properties of the Peer Support Survey. Sixth, participating PSP may not have had sufficient practice or time to broadly and successfully engage with the peer support skills or coping strategies. Future researchers should consider longer follow-up periods, more frequent measurement, and evaluating the impact of periodic skill refreshment.

## 5. Conclusions

The current study investigated the perceptions and impact of CISM peer support among integrated firefighters and paramedics in Canada. Mental disorders (i.e., alcohol use disorder, generalized anxiety disorder, major depressive disorder, panic disorder, PTSD, social anxiety disorder) are prevalent among PSP irrespective of individual differences (i.e., gender, age, years of service). Participants perceived their experience of CISM peer support as beneficial. Participants also perceived CISM as providing beneficial and valuable tools including skills (e.g., mindfulness) and coping strategies (e.g., tactical breathing) for assisting PSP and their peers with maintaining positive mental health. CISM may offer some specific mental health benefits relative to symptoms of alcohol use disorder and generalized anxiety disorder. The results continue to highlight the importance of registered mental health care providers among peer support teams as well as clear standard operating procedures for stepped-care referrals (e.g., psychotherapy). Overall, the current results offer novel information that can inform future directions for evidence-based peer support and policy decisions.

## Figures and Tables

**Table 1 ijerph-19-04976-t001:** Individual differences and mental disorders based on different CISM environments.

	All	CISM with High Fidelity	CISM with Unknown Fidelity	No
				CISM
	% (*N*)	% (*n*)	% (*n*)	% (*n*)
Number of Participants	215	91	60	64
Gender				
Men	86.0 (185)	93.3 (84)	91.7 (55)	71.9 (46)
Women	8.4 (18)	6.7 (6)	8.3 (5)	10.9 (7)
Age				
<30	11.2 (24)	14.3 (13)	8.3 (5)	9.4 (6)
30–39	39.5 (85)	50.5 (46)	31.7 (19)	31.3 (20)
40–50	28.8 (62)	29.7 (27)	28.3 (17)	28.1 (18)
>50	15.8 (34)	5.4 (5)	30.0 (18)	17.2 (11)
Years of Service				
<3	—	—	—	—
3–9	21.4 (46)	33.0 (30)	—	18.8 (12)
10–19	41.9 (90)	45.1 (41)	43.3 (26)	35.9 (23)
20–30	24.2 (52)	15.3 (14)	33.3 (20)	28.1 (18)
>30	6.5 (14)	3.3 (3)	15.0 (9)	17.2 (11)
Mental Disorders				
Any	36.7 (79)	38.5 (35)	36.7 (22)	34.4 (22)
Alcohol Use Disorder	17.2 (37)	12.1 (11)	26.7 (16)	15.6 (10)
Generalized Anxiety Disorder	10.2 (22)	8.8 (8)	8.3 (5)	14.1 (9)
Major Depressive Disorder	8.8 (19)	11.0 (10)	—	10.9 (7)
Panic Disorder	9.3 (20)	16.5 (15)	—	—
PTSD	9.3 (20)	8.8 (8)	8.3 (5)	10.9 (7)
Social Anxiety Disorder	9.8 (21)	11.0 (10)	—	10.9 (7)

Note: — *=* Not presented because of insufficient sample size (i.e., *n* < 5).

## Data Availability

The data presented in this study are available on request from the corresponding author.

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
