# Peer review of "Assessing the Perceptions and Impact of Critical Incident Stress Management Peer Support among Firefighters and Paramedics in Canada"

_ijerph, 2022, doi:10.3390/ijerph19094976_

Round 1

Reviewer 1 Report

Dear authors

Thank you very much for the very good manuscript on this exciting topic. It was a pleasure to read and review this manuscript. The structure of the manuscript is logical and very easy to follow.  The research question and the presentation of the hypotheses are clear and specific. The description of material and methods is transparent. The chosen scales are very well presented and described in brief. With regard to the paragraph on statistical analysis (p. 6), the authors should check whether the paragraph (lines 256-258) is intended in this form. It seems that rather format specifications have not been deleted here. The presentation of the results is transparent and comprehensible. The tables used are well integrated into the text. The discussion is comprehensive. The limitations of the study are critically described. The conclusion is concise and coherent. The authors could have additionally used the criteria of the EQUATOR network (e.g. STROBE or CONSORT) as a guideline for conducting the study and preparing the manuscript and referred to them.

Author Response

Thank you for the kind feedback! We have deleted the unintended sentences in the statistical analyses section as pointed out by the reviewer. 

Reviewer 2 Report

In general, the manuscript is in good quality and significant. Still, I have some suggestions for the authors. 

  1. Given that the assessment of the intervention has divided the participants into 4 groups, with high, low and unknown fidelity CISM, as well as control. I would suggest the authors to provide more information about the details or differences between high, low, and unknown fidelity of CISM, and the possible effect towards people mental health.
  2. Although your CISM experiment focuses on peer support, your major findings have been strongly related to mental disorders, which I think your topic should have included in order to provide the readers a clear picture while reading the topic and abstract. So as your research design as a quasi-experimental research. The abstract should also include more details about the 2 groups, and information of the participants. 
  3. I do agree with the use of regression in showing the effect of CISM, as well as controlling the mental disorder of the participants while examining. However, it will be more interesting to include a group differences between the experimental and control group on their mental issue levels after engaging in the CISM. By not just using Binary logistic regress, with only positive or negative. 

Author Response

Thank you for the feedback!

  1. To clarify, the study has three (not four) groups: a) CISM with high fidelity, b) CISM with unknown fidelity, and c) no CISM. We have revised both the abstract (lines 22-23), research questions and hypotheses (lines 114-115), and results (line 270) to make this more clear. As requested, we added more information regarding the term and concept of fidelity to the introduction (lines 96-98). We also added clarification that there is currently no research on the associations of fidelity on mental health (line 103).
  2. We agree that mental disorders is a strong topic throughout the manuscript. As such, we have updated the keywords to reflect his topic (line 30). We also added more information about the groups (lines 22-23) and participants (line 20) to the abstract.
  3. We agree that an additional analysis could be conducted to show the associations between groups and mental health symptoms in addition to the current analyses which show the associations between groups and mental disorder screens. Unfortunately, when we aggregated the data the total scores for the measures were removed from the dataset and lost. We apologize for this error and will ensure not to make this error in the future. 

Round 2

Reviewer 2 Report

I accept the authors' explanation towards my previous feedback and the revised content.